# Motor Intentions Decoded from fMRI Signals

**DOI:** 10.3390/brainsci14070643

**Published:** 2024-06-26

**Authors:** Sergio Ruiz, Sangkyun Lee, Josue Luiz Dalboni da Rocha, Ander Ramos-Murguialday, Emanuele Pasqualotto, Ernesto Soares, Eliana García, Eberhard Fetz, Niels Birbaumer, Ranganatha Sitaram

**Affiliations:** 1Psychiatry Department, Interventional Psychiatric Unit, Interdisciplinary Center for Neurosciences, Medicine School, Pontificia Universidad Católica de Chile, Santiago 8320165, Chile; sruiz@uc.cl; 2Laboratory for Brain—Machine Interfaces and Neuromodulation, Pontificia Universidad Católica de Chile, Santiago 8320165, Chile; 3Independent Researcher, Newton, MA 02459, USA; sangkyun@gmail.com; 4Department of Diagnostic Imaging, St. Jude Children’s Research Hospital, Memphis, TN 38105, USA; josueluiz.dalbonidarocha@stjude.org; 5Institute of Medical and Behavioral Neurobiology, University of Tubingen, 72076 Tübingen, Germany; ander.ramos@gmail.com; 6TECNALIA Basque Research and Technology Alliance (BRTA), 20009 San Sebastian, Spain; 7Department of Neurology & Stroke, University of Tubingen, 72074 Tübingen, Germany; 8Athenea Neuroclinics, 20014 San Sebastian, Spain; 9Independent Researcher, 1200 Bruxelles, Belgium; emanuele@pasqualotto.me; 10Coimbra Institute for Biomedical Imaging and Translational Research (CIBIT), University of Coimbra, 3000-548 Coimbra, Portugal; ernesto.soares@gmail.com; 11Bayer AG, 13342 Berlin, Germany; elilife@gmail.com; 12Departments of Physiology and Biophysics and DXARTS, Washington National Primate Research Center, University of Washington, Seattle, WA 98195, USA; fetz@uw.edu; 13Dipartimento di Neuroscienze (DNS), Universita degli Studi di Padova, 35131 Padova, Italy; birbaumer@uni-tuebingen.de

**Keywords:** motor intention, fMRI, frontal lobe, parietal lobe, motor imaginary, neurorehabilitation, brain–computer interfaces

## Abstract

Motor intention is a high-level brain function related to planning for movement. Although studies have shown that motor intentions can be decoded from brain signals before movement execution, it is unclear whether intentions relating to mental imagery of movement can be decoded. Here, we investigated whether differences in spatial and temporal patterns of brain activation were elicited by intentions to perform different types of motor imagery and whether the patterns could be used by a multivariate pattern classifier to detect such differential intentions. The results showed that it is possible to decode intentions before the onset of different types of motor imagery from functional MR signals obtained from fronto-parietal brain regions, such as the premotor cortex and posterior parietal cortex, while controlling for eye movements and for muscular activity of the hands. These results highlight the critical role played by the aforementioned brain regions in covert motor intentions. Moreover, they have substantial implications for rehabilitating patients with motor disabilities.

## 1. Introduction

The perception of movements is based on sensory information that reaches the brain from peripheral somatic receptors and, in the case of intact vision, also from input from the eyes. These processes are known as proprioception [1] and visual feedback [2], respectively. However, this idea has been challenged by the proposal that our awareness of movement execution primarily arises from our initial “intention” to move. According to this proposal, the sensation of moving is associated with increased activity in the parietal (perceptual) and frontal (motor) regions of the brain [3]. Thus, the concept of movement intention is framed as a higher-order cognitive function associated with the initial stages of movement planning, and this may specify the body part involved, centered on the target and contingent upon the task requirements [4].

Experiments with functional imaging in humans have suggested that conscious intentions to perform reach-and-grasp movements towards objects can be predicted from brain signals shortly before the movements are actually executed [5,6]. It remains unclear whether the concept of movement intention can be extended to covert movements, such as motor imagery, and whether the intention to engage differentially in various types of movement imagery can be predicted based solely on brain activity. Decoding the neural bases of intentions for goal-directed actions is important not only for gaining a deeper understanding of higher-level brain functions and awareness but also for developing innovative treatments for movement disorders. These treatments may include neurorehabilitation, non-invasive brain stimulation, brain–machine interfaces, and neuroprosthetics [7].

In the present study, we investigated whether differences in spatial and temporal patterns of brain activation, using blood oxygen level-dependent (BOLD) signals, were elicited by the participants’ intentions to perform different types of motor imagery (left vs. right hand) as already demonstrated for slow cortical potentials (SCPs) [8]. However, the poor spatial resolution of SCPs does not allow a clear definition of the anatomical–functional relationship between the direction of intention. We investigated whether a multivariate pattern classifier could use these patterns to identify movement intentions by using BOLD signals obtained from different brain regions. To this end, we designed an event-related functional magnetic resonance imaging (fMRI) paradigm. In this paradigm, participants were first prompted about the upcoming motor imagery they would perform (left or right hand) but were explicitly instructed to refrain from initiating the imagery until they received a further cue. To ensure that any observed brain activations were not influenced by hidden hand or eye movements of the participants, we monitored the muscle activity in their hands with electromyography (EMG) and tracked their eye movements during the experiment. We hypothesized that intentions for different types of motor imagery could be successfully decoded from BOLD signals of the brain, particularly from the parieto-frontal regions.

## 2. Materials and Methods

### 2.1. Participants 

Ten participants (five female, 21–26 years) participated in the study. All were free of any neurological or major disease or medication, and all had normal vision. They were right-handed as assessed by the Edinburgh Handedness Inventory [9]. All participants gave informed consent to participate in the study, which was approved by the local ethics committee of the Faculty of Medicine of the University of Tübingen, Germany.

### 2.2. Experimental Protocol

During the fMRI scanning sessions, the experimental protocol was presented visually to the participants via a mirror attached to the head coil and by using the Presentation software (Neurobehavioral Systems, Inc., Albany, CA, USA. Available online: https://www.neurobs.com/, accessed on 12 June 2024). The protocol consisted of an event-related design of successive runs composed of fixation, motor intention, and motor imagery blocks. Fixation block durations were pseudo-randomized to integral multiples of the MRI repetition time (TR) (1.5 s) between 1 and 5 TRs. The motor intention blocks had a duration of 1 TR, whereas the imagery blocks had a duration of 3 TRs (Figure 1). During the motor intention blocks, participants were presented with a left-pointing or right-pointing arrow at the center of the screen, indicating the direction of the upcoming motor imagery block (left or right hand). Participants were explicitly instructed not to initiate the imagery until they received further cues. During the subsequent imagery blocks, participants were instructed to perform kinesthetic motor imagery (involving imagining the sensation of performing a hand movement). The imagined movement involved a sequence of three sub-movements: first, reaching for an imagined tool placed approximately 10 cm in front of the hand; next, grabbing it; and finally, flexing the arm to lift the imagined object towards the ipsilateral shoulder. This sequence of movements was used because complex imagined movements produce stronger brain activations [10]. Participants mentally conducted these movements of the right or left hand according to the instructions presented. Each participant underwent four scanning sessions, with each session following an identical paradigm. One session consisted of 20 runs of fixation, motor intention, and motor imagery blocks for each hand, resulting in 40 trials in total. Left-hand and right-hand trials were pseudo-randomized.

Before the task, participants received detailed information about the protocol. Once placed in the scanner, participants underwent several practices run in the same position as was to be used during the experiment to familiarize them with the task.

### 2.3. FMRI Data Acquisition and Preprocessing

Experiments were conducted using a 3-Tesla MR Trio system (Siemens, Erlangen, Germany) with a standard 12-channel head coil. Functional image had 16 slices (voxel size = 3.3 mm × 3.3 mm × 5.0 mm, slice gap = 1 mm). Slices were AC/PC aligned in axial orientation. A standard echo-planar imaging (EPI) sequence was used (TR = 1.5 s, matrix size = 64 × 64, effective echo time TE = 30 ms, flip angle α = 70°, bandwidth = 1.954 kHz/pixel). For superimposing functional maps on brain anatomy, a high-resolution T1-weighted structural scan of the whole brain was acquired for each participant (MPRAGE, matrix size = 256 × 256, 160 partitions, 1 mm^3^ isotropic voxels, TR = 2300 ms, TE = 3.93 ms, α = 8°). Two foam cushions immobilized the participant’s head.

The preprocessing of the fMRI images was performed with SPM12 (the Wellcome Department of Imaging Neuroscience, London, UK), and classification was performed using MATLAB (Mathworks, Natick, MA, USA. Available online: https://la.mathworks.com/products/matlab-online.html, accessed on 12 June 2024) scripts. Realignment, co-registration, normalization onto the Montreal Neurological Institute space, smoothing (Gaussian kernel of 8 mm full width at half maximum), and whole brain masking were then performed. To account for the variance in the BOLD signals for different participants, z normalizations were applied across all the time series to the data for each participant data separately.

### 2.4. Support Vector Classification

To decode brain activations related to motor intentions, a multivariate analysis was performed using a machine learning algorithm called support vector machine (SVM) [11,12], a pattern recognition technique which has shown high performance in comparison to other existing methods of pattern classification of fMRI signals [13,14,15,16,17]. The SVM software SVMlight [18] was used to implement the classifier [11]. Linear kernel SVM, using leave-one-out cross-validation [19], was trained with a fixed regularization parameter C = 10^5^ to remove variability in classification performance dependent on the regularization parameter C.

For SVM analysis, pre-processed images were obtained with SPM5, and classification was performed using an in-house MATLAB (Mathworks, Natick, MA, USA. Available online: https://www.cs.cornell.edu/people/tj/svm_light/, accessed on 12 June 2024) toolbox [20]. For SVM classification, the previously computed z-values were used as features.

First, we examined whether it was possible to distinguish between successive conditions (fixation, intention, and imagery) by classifying the spatial patterns of brain signals extracted from brain regions of interest (ROIs) known to be involved in motor planning and execution, namely the posterior parietal cortex (PPC), supplementary motor area (SMA), premotor cortex (PMC), and primary motor cortex (M1). To explore the specificity of these regions coding for motor intention, the dorsolateral prefrontal cortex (DLPFC), a large brain region not usually considered to be involved in motor tasks, was included in the analysis. The posterior cingulate cortex [21] and fronto-polar cortex [22] were also included because of their involvement in the preparation for overt motor execution. The somatosensory area was included based on findings from neural recordings in monkeys, which demonstrated that neural activity in the postcentral cortex precedes active limb movement [23].

To evaluate the decoding accuracy from data for these brain areas, brain masks were created with WFU PickAtlas Toolbox (Available online: https://www.nitrc.org/projects/wfu_pickatlas, accessed on 12 June 2024) by using Brodmann areas (BAs) as follows: BA 10 for the frontopolar cortex, BA 9 + 45 + 46 for the DLPFC, the mesial part of BA 6 for the SMA, the remaining part of BA 6 for the PMC, BA 4 for the M1, BA 1 + 2 + 3 for the primary somatosensory cortex, BA 5 + 7 + 39 + 40 for the PPC, and BA 31 for the posterior cingulate cortex. To separate the mask of the PMC from the mask of BA6, the mask of the SMA (in the AAL labels of the WFU PickAtlas Toolbox) was subtracted from that of BA6. The classification performance from data was evaluated through 4-fold cross-validation [24]. The pattern analysis accounted for the delay in the hemodynamic response with respect to the stimulus onset by introducing an equivalent delay of 3 s (2 TRs) in the input data set [25].

### 2.5. Multivariate Spatial Analysis with Effect Mapping

Based on the parameters of the trained SVM model, we analyzed the fMRI data with effect mapping [11] (EM). To identify informative voxels from the SVM model, EM measures the effect of each voxel in multi-voxel space to the SVM output by considering two factors, namely, the input vectors and the weight vector; $y=w^{T}x+w_{0}$, where y is the SVM output, w is the weight vector, and x the input vector which determine the SVM output. The effect of each voxel on the classifier output is measured by computing normalized mutual information (NMI) between the voxel and the SVM output. MI is defined as the amount of information that one random variable contains about another random variable [26]. That is, when two random variables x and y occur with a joint probability mass function p(x,y) and marginal probability function p(x) and p(y), the entropies of the two random variables and the joint probability are given respectively by the following:(1)$$\begin{array}{c} H\left(X\right)=\sum_{x\in X}-p\left(x\right)\log p\left(x\right),\;H\left(Y\right)=\sum_{y\in Y}-p\left(y\right)\log p\left(y\right),\\ and\;H(X,Y)=\sum_{x\in X}\sum_{y\in Y}-p(x,y)\log p(x,y) \end{array}$$

MI, I(X;Y), is the relative entropy between the joint distribution and the product distribution, i.e.,
(2)I(X;Y)=H(X)+H(Y)−H(X,Y)

To correct for variance in mutual information based on entropies H(X) and H(Y), normalized mutual information is defined as [27]
(3)$$\tilde{I}(X;Y)=\frac{I(X;Y)}{H(X)+H(Y)}$$

Hence, the effect value (EV) $E_{k}$ of a voxel *k* is defined as
(4)$$E_{k}=w_{k}\tilde{I}(x_{k};y),\;k=1,\cdots ,M\;(M:\mathrm{number}\;\mathrm{of}\;\mathrm{voxels})$$
where y is the SVM output after excluding the sign function, and $w_{k}$ and $x_{k}$ are the SVM weight value and activation in voxel *k*, respectively. 

After normalizing the absolute value of $E_{k}$ from Equation (4), we obtain the following relation:(5)$$nE_{k}=\mathrm{sgn}(E_{k})\log\left(1+\left|E_{k}\right|/std(\left|E\right|)\right),k=1,\cdots ,M$$
where sgn(.) is a sign function, and $std(\left|E\right|)$ is the standard deviation of all $E_{k}$ values. In the present study, Equation (5) ($nE_{k}$) was used to compute the EV at each voxel to make E(effect)-maps from different participants and different folds of cross-validation comparable.

With different contrasts, i.e., intention vs. fixation and left vs. right over time points (left and right fixations, left and right intentions, and left and right imaginations), E-maps were separately obtained from data taken together from all the brain areas of the participants. The E-maps from a contrast for 4-fold CV of all participants (i.e., 40 E-maps; 4 E-maps from 4-fold CV and 10 participants) were averaged into an E-map for a group analysis, and then the averaged map was smoothed spatially with 5 mm fixed width at half maximum (FWHM) to minimize the distortion of the map for ease of interpretation. In the interpretation of the E-map, positive and negative EVs were related the design labels, 1 and −1, of the SVM classifier, respectively. That is, if the design labels of two conditions are exchanged, the sign of EVs are also reversed.

### 2.6. Eye Tracker

To ensure that classification of neural data corresponded to motor intentions, and not to “overt movements”, we collected and analyzed information coming from potential eye and hand movements. These analyses were performed to rule out the possibility that the classification accuracy obtained from brain data could be due to muscle activity potentially elicited during the visual stimuli of the experimental protocol (Figure 1).

For potential eye movements, we monitored pupil positions inside the scanner using EYE-TRAC^®^ 6 (Applied Science Laboratories, Bedford, MA, USA), a video-based infrared eye-tracker with long-range optics, specifically designed for fMRI, and a sampling rate of 60 Hz in 9 participants, simultaneously with a collection of the fMRI data. After the removal of blinks and outliers, we performed a discriminant analysis to find out whether the data could be classified according to the lateralization (left or right) of each condition (intention and imagery). Discriminant analysis [28] is a statistical method that can be used to develop a predictive model of group membership based on observed features of the data. Starting from a sample of cases with a known group membership, a discriminant function is generated based on the linear combination of the variables. This function, which provides the best discrimination between the groups, can be then applied to new cases. In the literature, three different types of discriminant analysis can be found: direct, hierarchical, and stepwise. The difference in these three methods consists of entering the variables for the function. In the direct method, the one we used, all variables are entered together; in the hierarchical one, the researcher determines the order; and in the stepwise discriminant analysis (SWDA), variables are entered step by step and are statistically evaluated to determine which one contributes the most to discriminate. In our analysis, the lateralization of the conditions was used to determine the discriminative function. We computed for the 9 participants (4 sessions per participant) a 10-fold cross-validated discriminant analysis for each TR. The results of this analysis provided us with means and standard errors.

### 2.7. Electromyography

In order to explore whether classification accuracies obtained from brain areas could have been influenced by muscular activity, participants were instructed to remain still and avoid any movement during the experiment. Furthermore, 6 participants performed the same experimental task again, in which EMG data were recorded simultaneously. The EMG data were not obtained in the same sessions in which brain data were analyzed in order to avoid participants becoming distracted from the task due to the presence of the EMG recording, to avoid brain activations due to skin stimulations by the electrodes and wires, and to avoid possible artifacts in the brain signals due to the EMG. EMG data were acquired using 6 pairs of bipolar, sintered Ag/AgCl electrodes, which were placed based on physical landmarks on antagonistic muscle pairs, with 3 pairs in each arm. One pair was placed close to the external epicondyle over the extensor digitorum (extension), the second pair over the flexor carpi radialis (flexion), and the last pair over the external head of the biceps (flexion). The electrode wires were twisted per pair to minimize the differential effect of the magnetic field on the EMG leads. A ground electrode was placed on the ankle joint. Current-limiting resistors (5 kΩ) were attached to the EMG electrodes to prevent the possible warming of the electrodes. All electrodes were connected to an electrode input box, which was in turn connected to the amplifier. The digital signals were transmitted via an optical cable and stored on a personal computer outside the MR room. Data were recorded using a MR-compatible bipolar 16-channel amplifier (BrainAmp) from Brain Products GmbH, Munich, Germany. The sampling rate of data acquisition was set at 5000 Hz with a low-pass filter of 250 Hz and a signal resolution of 0.5 muV. The acquired EMG signal was synchronized with the scanner clock using the SynchBox device from Brain Products GmbH, Munich, Germany. The SyncBox scanner interface serves as the direct receiver for pulses from the MR gradient clock (10,000 kHz). A model of the MR gradient artifact was generated by averaging EMG signals from five repetition times (TRs, 1.5 s) of the echo-planar imaging (EPI) pulse sequence. The MR artifact template obtained was then subtracted from the original data for correcting the gradient artifacts. The artifact-corrected and -filtered data were subsequently used for pattern classification in the following manner. EMG signals for each hand were separated into data sets for distinct motor class labels as follows: left and right intention trials and left and right imagery trials. Here, each trial corresponds to EMG data acquired during one complete TR of the EPI pulse sequence. The classes, left and right intention, corresponded to 1 TR, and the classes left and right imagery to 3 TRs, respectively, of the EMG data.

The complexity of the EMG waveform for each TR (1.5 s) was determined as a time-domain feature from a moving window of 240 ms and a window overlap of 24 ms. The waveform was computed from the following equation:(6)$$WL=\sum_{k=1}^{L}\left|\Delta x_{k}\right|\;\Delta x_{k}=x_{k}-x_{k-1}$$
where x is the rectified EMG data of the window from data point k=1 to k=L, with L being the length of the window. The feature, WL, of the signal is a combined indicator of signal amplitude and frequency. The extracted feature was then transformed to a principal component space by performing a discrete Karhunen–Love transform. Non-linear decoding filters were designed using multilayer, feed-forward artificial neural networks (ANNs) because of their use in nonlinear regression and classification. By using a tan-sigmoid transfer function for the hidden layer neurons and a log-sigmoid transfer function for the output layer, the network assigns a probability to each movement, P{Mi}, where i=1, and 2 corresponds to the 2 hand movement types (right and left). The movement type with the highest probability is chosen as the final output of the classifier. The neural network was trained using MATLAB’s scaled conjugate gradient descent algorithm in combination with early validation to improve generalization. The results were validated using a 10-fold cross-validation technique.

## 3. Results

First, we examined whether it was possible to distinguish between successive conditions (fixation, intention, and imagery) by classifying the spatial patterns of brain signals. The results of multivariate pattern analysis using a SVM showed that the classification accuracies for all these areas were above chance (>50%) for classification between blocks of fixation and motor intention and between motor intention and motor imagery. Furthermore, the PMC and PPC displayed the highest classification accuracies (>80%) when compared with other ROIs (see Table 1). Based on the parameters of the trained SVM model, we further analyzed the fMRI data with the EM method of multivariate functional analysis [11]. The E-maps (Figure 2) show a clear distinction between the activation patterns for the different conditions and relative differences between the ROIs in intention formation and imagery, thus confirming the presence of discriminative information in the PPC, PMC, and SMA (for Figure 2 and Table 1, classification was performed averaging the bold values for each voxel to one data point for each condition, i.e., fixation and imagery. Classification accuracies were averaged across all the participants).

In the second step, we further investigated the specificity of the information in the ROIs by testing whether the pattern classifier could decode the laterality (left or right) of the task. In the motor intention block, the classifier could robustly distinguish between left and right tasks in the PPC (accuracy = 63.1%) and in the PMC (accuracy = 61.9%), showing that information about movement intention direction is available in these areas (Table 2). During the imagery blocks, classification accuracies of >80% distinguished left imagery from right imagery in the PMC, PPC, and SMA. Again, particularly high classification results were obtained for the PPC and PMC (mean = 86%) (Table 2 and Figure 3A). The E-maps, displaying the most informative voxels for the classification between left and right intention, confirm the important roles played by the PPC and PMC in intention formation (Figure 3B).

To confirm that these results were not influenced by physiological factors such as eye movements and hand muscle activity, we acquired eye tracker and EMG signals within the fMRI sessions and performed pattern classifications of these signals by using linear discriminant analyses and neural network analyses. The results of the classification of pupil positions for left vs. right motor intention, as well as for motor imagery, were around the chance level. Similar results were obtained by classifying the EMG signals of the left and right hands (Table 2 and Figure 3A). As intentions for movement are shown to be anatomically segregated in the PPC, with regions being specialized for planning saccades, in addition to reaches and grasps, these results show that the fMRI classification is specifically derived from movement imagery and is not related to systematic eye movements. 

## 4. Discussion

Our results show that intentions for different types of motor imagery can be predicted from BOLD signals in the parieto-frontal regions (particularly the PPC and PMC) of the human brain with a support vector classifier. Varied evidence shows that the PPC plays a crucial role in movement planning [29] and that it contains anatomically segregated regions (intentional maps) that code for the planning of different movements [30]. Furthermore, electrical stimulation of the inferior parietal cortex in human patients with brain tumors caused a strong intention and desire to move, whereas more intense stimulation of this area led to illusory movement awareness, thus lending credence to the hypothesis that both motor intention and motor awareness emerge from activations of parietal regions [31]. Having said that, the PMC has anatomical connections with the frontal, parietal, and motor cortical regions [32]. This establishes the PMC as a central hub in the processes of motor planning and execution, effectively transmitting information regarding the advanced cognitive functions associated with movement [33]. The high classification accuracies obtained in the present study during the intention block for the PPC and PMC suggest that these regions play analogous roles in intention formation for both overt and covert movements.

The specificity of these results could be seen in light of the comparatively lower classification accuracies obtained from the other brain areas included in the analysis. This point is further reinforced by considering the almost chance classification accuracy obtained from the DLPFC for both intention and imagery, despite this being a relatively large region in the brain.

Interestingly, for left vs. right classification during both movement intention and imagery, a general tendency for higher prediction accuracies for the PPC and PMC, compared to that for M1, was found across every time point. These findings are concordant with the idea that premotor and parietal regions play a predominant role in different aspects of action planning [34,35,36] and that the primary motor cortex is inconsistently activated during motor imagery, usually at a lower intensity than during motor execution [37].

It might be argued that the experimental paradigm did not ensure that participants did not perform mental imagery during the intention block. However, this interpretation is not tenable in view of the high classification results for differentiating between motor intention and motor imagery and the different brain areas involved in the two conditions (see Table 1 and Figure 2). It might be also argued that the high classification accuracies obtained for intention were merely the result of a nonspecific expectancy for the forthcoming go signal or imagery block. However, in this case, it would have been impossible for the classifier to distinguish between left and right motor intentions (see Table 2 and Figure 3).

We propose that the concept of intention that was studied in this experiment belongs to the class of “immediate intentions”, which are accompanied by conscious experiences of impending actions [38]. Immediate intentions can be distinguished from prospective intentions based on how early the episodic details of an action are planned. In the former case, the time lag between an intention and its action can be very short and may not even be consciously separable. In the latter case, the time difference between the formation of an intention and its actual execution may be quite prolonged, as in the planning of a holiday today and its actual execution several days later. It has been proposed that immediate intentions have a feature that makes a clear prediction about the oncoming action. Termed the content argument, this states that intentions for two different actions (e.g., left vs. right hand) have two different contents in the brain, capable of explaining or predicting which body part will be used for movement. In our study, we have shown evidence for the content hypothesis by predicting with high accuracy left movement imagery (of reaching and grasping), distinct from right movement imagery, by using fMRI signals before the onset of imagery.

Our results suggest future applications in which high-level preparatory activity in the parieto-frontal regions can be applied to control neural prosthetics in BCI. We assume that repeatable formations of motor intentions may persist in the absence of overt movement in those patients who can maintain or be trained to maintain mental simulations of movement, a concept that has important implications for the rehabilitation and restoration of movement, e.g., after a stroke. In this rehabilitation setting, predicting movement intention could provide a sufficient time advantage for triggering the activation of somatosensory feedback through devices such as functional electric stimulation or robotic orthoses, promoting neural plasticity [39]. 

Also, if a paradigm similar to that used here is implemented, a portable brain–computer interface (BCI) based on EEG [40] or near-infrared spectroscopy (NIRS) [41] could be built to operate a neuro-prosthetic device as an “intention brain–machine interface”. There is an advantage in using higher, cognitive areas of the sensorimotor system, in particular the PPC, for the following reasons: (1) although the motor regions may undergo degradation during paralysis, the PPC may suffer less degradation because of its close link to the visual system, which is still intact; (2) as movement restoration is accompanied by neural plasticity, it is hypothesized that the PPC plays a larger role in this recovery because of its involvement in the sensorimotor re-registration of behavior; and (3) for an accurate control of movement, closed-loop feedback is necessary, and this is largely lost in the motor areas as a result of lesions but often remains intact in the PPC because the re-afference to this region is also visual.

## Figures and Tables

**Figure 1 brainsci-14-00643-f001:**
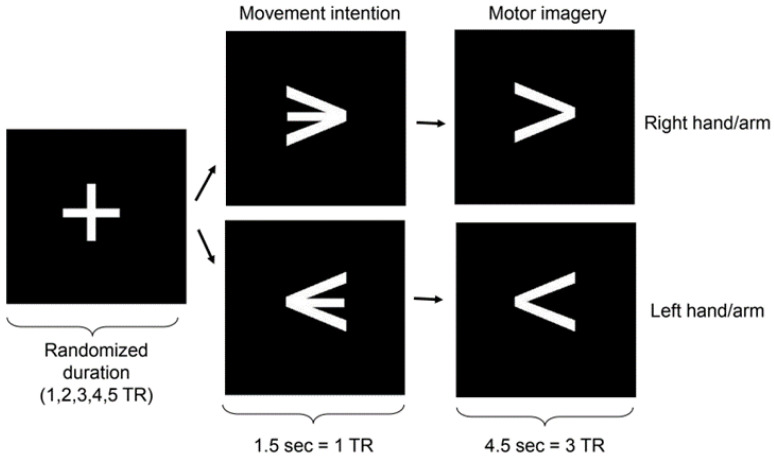
Stimulus presentation paradigm. The figure shows the blocks of fixation, motor intention, and motor imagery (TR = repetition time).

**Figure 2 brainsci-14-00643-f002:**
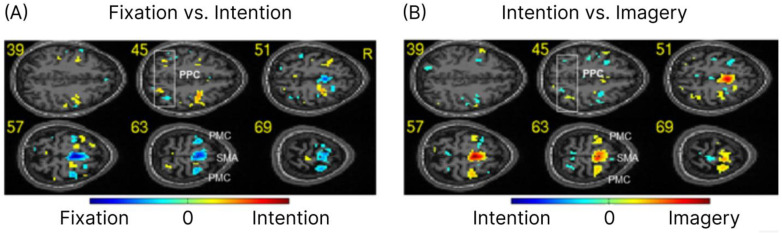
Activation maps for classification between successive conditions. (**A**) Effect maps (E-maps) for the classification between fixation and motor intention. (**B**) E-maps for the classification between motor intention and motor imagery. For display purposes, the E-maps were drawn by selecting the more informative voxels (top 20% of the voxels with the highest effect values). The figure shows six horizontal slices of the brain at spatial intervals of 6 mm in Montreal Neurological Institute (MNI) coordinates (numbers represent z-coordinates). R: right; PMC: premotor cortex; SMA: supplementary motor area; PPC: posterior parietal cortex.

**Figure 3 brainsci-14-00643-f003:**
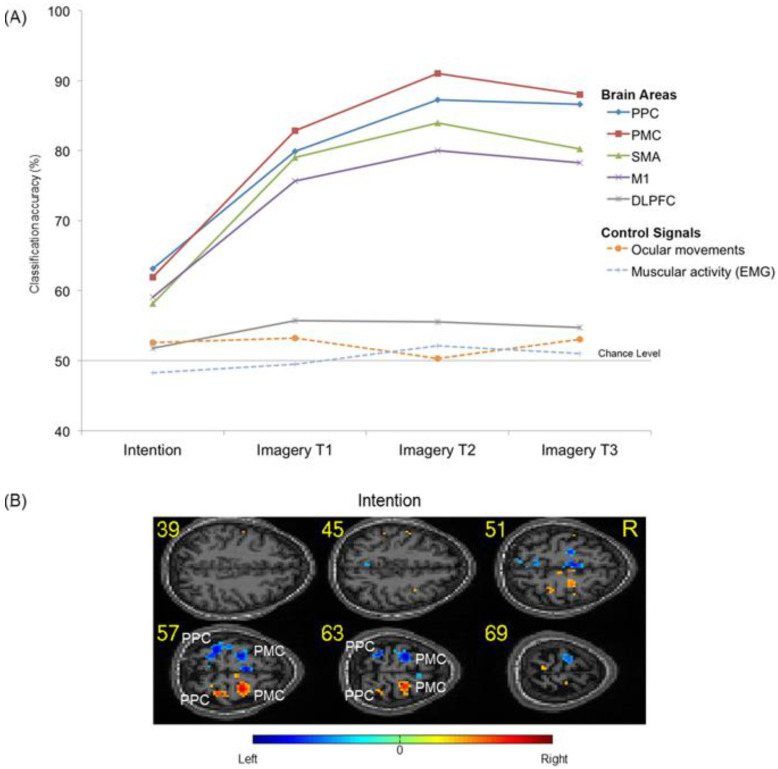
Decoding accuracy and activation maps for left vs. right classification. (**A**) Decoding accuracies for “left vs. right” across the conditions of intention (1 TR) and motor imagery (3 TRs: imagery T1, T2, and T3). For the sake of clarity, only data from the PPC, PMC, SMA, M1, DLPFC, ocular movements, and muscular activity are displayed (for details, see Table 2). (**B**) Activation maps for left vs. right during intention. For display purposes, the activation maps were drawn by selecting the top 20% most discriminating voxels. Numbers represent z-coordinates in the Montreal Neurological Institute (MNI) system. PMC: premotor cortex; SMA: supplementary motor area; PPC: posterior parietal cortex; DLPFC: dorsolateral prefrontal cortex.

**Table 1 brainsci-14-00643-t001:** Classification accuracies (and standard errors of the mean) for fixation vs. intention and for intention vs. imagery. The table shows the classification accuracies of multivariate pattern analysis across successive conditions for different ROIs. PMC: premotor cortex; PPC: posterior parietal cortex; SMA: supplementary motor area; DLPFC: dorsolateral prefrontal cortex.

	Fixation vs. Intention		Intention vs. Imagery	
	Mean	SE	Mean	SE
PMC	83.9%	1.4	85.9%	1.6
PPC	82.8%	1.4	83.3%	1.6
SMA	77.4%	1.5	80.7%	1.8
M1	72.7%	1.5	74.3%	1.7
Posterior cingulate	70.2%	1.1	70.5%	1
DLPFC	74.8%	1.2	76.3%	1.3
Somatosensory area	72.9%	1.5	75.2%	1.6
Frontopolar cortex	62.4%	1.3	61.3%	1.3

**Table 2 brainsci-14-00643-t002:** Classification accuracies (and standard errors of the mean) for brain areas and control conditions (ocular moment and muscular activity) for “left vs. right” during motor intention (1 TR) and motor imagery (3 TRs: T1, T2, T3). PMC: premotor cortex; PPC: posterior parietal cortex; SMA: supplementary motor area; DLPFC: dorsolateral prefrontal cortex.

	Intention		Imagery					
			T1		T2		T3	
	Mean	SE	Mean	SE	Mean	SE	Mean	SE
PPC	63.1%	1.2	79.9%	1.3	87%	1.3	86.6%	1.6
PMC	61.9%	1.2	82.9%	1.7	91%	1.4	88%	1.7
SMA	58.1%	1.4	79%	1.4	84%	2	80.2%	2.2
Somatosensory area	58.1%	1.7	78.1%	1.9	82.5%	2.1	82.8%	2
M1	59.1%	1.5	75.7%	2.4	80%	2.4	78.3%	2.5
Posterior cingulate	52.8%	1	63.4%	1.2	69.6%	1.2	66.8%	1.8
DLPFC	51.8%	1	55.7%	1.1	55.5%	1.2	54.7%	1
Frontopolar cortex	51.8%	1	49.3%	1.1	50.9%	1.2	54.2%	1.3
Ocular movements	52.6%	0.1	53.1%	0.1	50.3%	0.3	53.5%	0.2
Muscular activity	48.2%	1.7	49.5%	1.3	52.1%	1.1	51%	1.4

## Data Availability

The datasets presented in this article are not readily available because are part of an ongoing study or due to technical/time limitations. Requests to access the datasets should be directed to: ranganatha.sitaram@stjude.org.

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
