# Peer review of "Motor Intentions Decoded from fMRI Signals"

_brainsci, 2024, doi:10.3390/brainsci14070643_

Round 1

Reviewer 1 Report

Comments and Suggestions for Authors

Details: Although the design seems to be apparent to the readers, the author should elaborate on the aspects of the control measures for eye and hand movements. In particular, the writer may clarify how these measures make sure that the activity of the brain recorded is directly connected to motor intention, not peripheral movements. The manuscript also lacks information on statistical methods. More details on the training and validation of the SVM classifier should be provided, as well as how the author handled imbalanced data sets. Details on the kernel type are also missing. More explicit connection between the observed activations in the brain can be provided with the actual understanding of motor intentions.

Author Response

  1. Although the design seems to be apparent to the readers, the author should elaborate on the aspects of the control measures for eye and hand movements. In particular, the writer may clarify how these measures make sure that the activity of the brain recorded is directly connected to motor intention, not peripheral movements.

Response: We thank the reviewer for this suggestion

To make this methodology clearer, we have added the following paragraph to the manuscript

“To ensure that classification of neural data correspond to motor intentions, and not to “overt movements”, we collected and analyzed information coming from potential eye and hand movements. These analyses were performed to rule out the possibility that the classification accuracy obtained from brain data could be due to muscle activity, potentially elicited during the visual stimuli of the experimental protocol (figure 1)

  1. The manuscript also lacks information on statistical methods. More details on the training and validation of the SVM classifier should be provided, as well as how the author handled imbalanced data sets. Details on the kernel type are also missing. More explicit connection between the observed activations in the brain can be provided with the actual understanding of motor intentions.

Response:

We thank the reviewer for this suggestion

Now, the following statement was included in the article (with corresponding new references -in the text-)

 “The SVM software SVMlight was used to implement the classifier. Linear Kernel SVM, using leave-one-out cross-validation, were trained with a fixed regularization parameter C = 105 to remove variability of classification performance dependent on the regularization parameter C.”

Regarding the balance of the data, for Figure 2 and table 1 classification was performed averaging the bold values for each voxel to one data point for each condition, i.e., fixation, and imagery. For Figure 3 there was no need to balance the data as the classification was made for every individual time point (TR) of the protocol. We have now added this explanation in our manuscript.

Reviewer 2 Report

Comments and Suggestions for Authors

The study outlined in this manuscript aimed to decoding movement intentions before motor imagery using fMRI signals. This is meaningful as it was only confirmed that the intentions before movement execution could be decoded. The findings in this study could be useful for the building of brain computer interface systems. Over all, the design of the experiments and data processing were reasonable, and the manuscript was well written. However, revisions are still needed for the manuscript to meet the requirements for publishing.

Line 133, Page 4. I have critical opinions on the statement about the superior performance of SVM on fMRI signals.

What is the type of the kernel used in your SVM analysis? Every study using SVM should report this setting. And please confirm you have compared the results of using different kernels.

Besides SVM, the results using linear discriminant analysis (LDA) should also be reported, since it will give an insight about the nature of the data.

Some clarifications for the results in Table 1, Figure 3A and Table 2 are needed. Were the shown numbers averaged across multiple sessions of one participant, or all the 10 participants?

There were many misalignment between the equations and texts in Section 2.5. Please work with the editors to fix them. The qualities of the figures in Figure 2 and Figure 3B could be improved, like the resolutions and font sizes.

Comments on the Quality of English Language

Some texts could be more concise, like those in Section 2.1.

Author Response

Line 133, Page 4. I have critical opinions on the statement about the superior performance of SVM on fMRI signals.

What is the type of the kernel used in your SVM analysis? Every study using SVM should report this setting. And please confirm you have compared the results of using different kernels.

Response:
We thank the reviwer for these suggestions:
A linear kernel was used for SVM classification (we have now added this information in our manuscript)

We did not compare systematically the results with different kernels as the chief purpose of the study was to use the classifier‘s weight values to map the discriminating voxels back on to the brain for neuroscience interpretation, for which the linear kernel is best suited. The use of other kernels including non-linear kernels would have made brain mapping and interpretation not straight forward.

Besides SVM, the results using linear discriminant analysis (LDA) should also be reported, since it will give an insight about the nature of the data.

Response:

We thank the reviewer for noticing a potential source of confusion in our manuscript

We did not performer LDA on the neural information (but SVM). For eye tracker data, we used discriminant analyses; and for the EMG data, neuronal network analyses. We have now clarified these points through the text

Some clarifications for the results in Table 1, Figure 3A and Table 2 are needed. Were the shown numbers averaged across multiple sessions of one participant, or all the 10 participants?

Response: We have now clarified in the text that the shown numbers were averaged across all 10 participants

There were many misalignment between the equations and texts in Section 2.5. Please work with the editors to fix them. The qualities of the figures in Figure 2 and Figure 3B could be improved, like the resolutions and font sizes.

Response:

We thank the reviewer for noticing this problems. Now, in accordance the Brain Sciences “Instructions for Authors”, the Equations are configured into Microsoft Equation Editor 3.0. They are now editable by the editorial office and do not appear in picture format anymore. We believe this have solved the misalignment problem

Also, we have changed the font sizes in Figure 2 and Figure 3B to increase the quality of the text resolutions (we have now included the new figures in the text)

Round 2

Reviewer 2 Report

Comments and Suggestions for Authors

I appreciate the authors’ efforts to improve the manuscript. It is notable that both texts and figures were significantly improved and my previous concerns were addressed by the new results in the revised version.